# Telephone versus In-Person Pharmacist-Led Medication Reviews in Home Dialysis Patients: Evaluating Quality of Care and Patient Satisfaction

**DOI:** 10.3390/pharmacy11010001

**Published:** 2022-12-21

**Authors:** Kelsey Allen, Andrew J. Flewelling, Lauren Munro, Heather Naylor

**Affiliations:** 1Department of Pharmacy, Nova Scotia Health, Aberdeen Hospital, New Glasgow, NS B2H 3S6, Canada; 2Department of Research Services, Horizon Health Network, Hilyard Place, 560 Main St., Suite A-200, Saint John, NB E2K 1J5, Canada; 3BioScript Pharmacy Ltd., 105-10 Ragged Lake Blvd, Halifax, NS B3S 1C2, Canada; 4Department of Pharmacy, Horizon Health Network, Saint John Regional Hospital, 400 University Ave, Saint John, NB E2L 4L2, Canada

**Keywords:** telehealth, pharmacy, dialysis, chronic kidney disease, medication reviews

## Abstract

The COVID-19 pandemic required pharmacists in a provincial Home Dialysis Clinic to adapt from in-person to telephone-based medication reviews. Studies have shown that in-person pharmacist interventions in patients with chronic kidney disease (CKD) lead to a reduction of drug therapy problems (DTPs), however, it’s unknown if telephone interventions provide similar outcomes. The purpose of this study was to evaluate whether differences in quality of care exist between in-person vs. telephone medication reviews in home dialysis patients and to evaluate patient satisfaction with telephone medication reviews. Data from the two most recent in-person medication reviews was compared with the two most recent telephone medication reviews for each patient (*n* = 46). There were no statistically significant differences in DTPs identified between in-person and telephone medication reviews (*p* = 0.431). Physician acceptance of pharmacist recommendations was higher for in-person medication reviews (*p* = 0.009). Patients were satisfied with the care they received with pharmacist-led telephone medication reviews, however, 29% (*n* = 7) would prefer an in-person medication review once per year with telephone medication reviews the rest of the time. Overall, patients were satisfied with the care they received from telephone medication reviews.

## 1. Introduction

Patients with end stage kidney disease undergoing dialysis are at a high risk of experiencing adverse drug events and other negative outcomes due to polypharmacy and multiple comorbidities [1]. Dialysis patients have the highest pill burden of all chronically ill patient populations with an estimated daily average of 12 medications [2]. Consequently, they are at extremely high risk of experiencing drug therapy problems (DTPs), defined as an actual or potential undesirable incident related to medication that affects the goals of therapy [3]. Integrating pharmacists into the care of hemodialysis patients has been shown to reduce mortality, medication use and length of hospitalizations [4].

The COVID-19 pandemic introduced unprecedented challenges to direct patient care in the ambulatory setting. Telepharmacy gained popularity as a method of supporting pharmacist medication management while maintaining social distancing [5]. A pre-COVID systematic review of 34 studies demonstrated that clinical pharmacy telemedicine interventions in the outpatient or ambulatory setting, primarily via phone, had an overall positive impact on chronic disease management [6], a finding corroborated by a systematic review of inpatient intensive care unit (ICU) and non-ICU telepharmacy services published in the same year [7]. The evidence supporting telepharmacy in the ambulatory care setting has continued to grow since the COVID-19 pandemic, with studies showing benefit of telepharmacy interventions in chronic diseases such as hypertension [8], diabetes [9,10] and in the management of anticoagulation medications [11]. Unfortunately, there are no published studies of telepharmacy interventions in chronic kidney disease (CKD) or dialysis patients to date. Studies have shown that in-person pharmacist medication reviews in patients with chronic kidney disease (CKD) lead to a reduction of DTPs and improvement in management of chronic diseases, such as anemia, diabetes, hypertension and dyslipidemia [12,13]; however, it is unknown if there is a similar benefit with telephone-based medication reviews [13]. 

The Home Dialysis Clinic located at the Saint John Regional Hospital (Saint John, NB, Canada) provides care to home dialysis patients (peritoneal dialysis and home hemodialysis) within New Brunswick’s Horizon Health Network. Restrictions arising from the COVID-19 pandemic required a transition from in-person clinic visits to telephone clinic visits for home dialysis patients. Pharmacists therefore had to adapt patient care practices in order to transition from in-person medication reviews to telephone medication reviews. 

Based on the positive outcomes seen with telephone-based pharmacy interventions in other chronic disease populations, we hypothesized that similar outcomes would be observed with telephone medication reviews versus in-person medication reviews in home dialysis patients. 

The purpose of this study was therefore to evaluate whether differences in quality of care existed between in-person pharmacist medication reviews versus telephone pharmacist medication reviews for home dialysis patients. 

## 2. Materials and Methods

### 2.1. Study Design

A within-subjects, retrospective study with a participant satisfaction survey was conducted at the Saint John Regional Hospital (SJRH) in Saint John, New Brunswick, Canada. This project was approved by the Horizon Health Network Human Research Protection Program on 5 January 2021 (file number 101134).

### 2.2. Participants

All adult patients (≥19 years of age) followed by the home dialysis clinic at the Saint John Regional Hospital (Saint John, NB, Canada) on an outpatient basis for home dialysis (either home hemodialysis or peritoneal dialysis) during the period of data collection were included. Patients were required to have at least two scheduled in-person medication reviews prior to March 2020 and two scheduled telephone medication reviews after March 2020 with the clinical dialysis pharmacist. 

### 2.3. Pharmacist Medication Reviews

All in-person and telephone medication reviews were completed by the same core group of three pharmacists. All pharmacists had been working in the home dialysis clinic for greater than 5 years prior to study onset. Medication reviews were performed four times per year during both the in-person and telephone study periods. During medication reviews, the pharmacists would review relevant lab values, complete a best possible medication history, identify and resolve DTPs, provide patient education, and address patient concerns. The same standardized patient assessment form was used for in-person and telephone medication reviews, however, the documentation format differed as the switch to telephone medication reviews necessitated a transition from paper to electronic documentation. No formal training was provided following the switch to telephone medication reviews as COVID-19 necessitated an urgent turnaround time to address patient care gaps. Pharmacists relied on cumulative experience and regular team meetings to adapt their practice for telephone care based. 

### 2.4. Data Collection

#### 2.4.1. Quantitative Data

Patients that met inclusion criteria were assessed by the primary investigator with the following baseline demographics recorded: age, sex, dialysis modality, length of time on dialysis, comorbidities, and number of medications. Data was collected for each patient from two in-person medication reviews and two telephone medication reviews. Data from the most recent telephone medication reviews was used to mitigate possible confounders during the immediate transition period from in-person to telephone. For in-person medication reviews, data was collected from paper charts and for telephone medication reviews from an electronic charting system (Renal Insight). The number of recommendations as well as the number and type of drug therapy problems were identified and collected from pharmacist documentation in paper and electronic patient charts. Physician acceptance of pharmacist interventions was identified and collected from clinic notes and medication orders in patient charts. Data collection was performed by a pharmacist who was not a member of the medication review team (KA).

#### 2.4.2. Qualitative Data

A survey consisting of closed and open-ended questions was distributed electronically using the Opinio survey software. The survey was distributed to eligible home dialysis patients and responses were collected using the online software. The primary investigator (KA) contacted all eligible patients to inform them of the survey and provided the option to complete the survey online or over the phone. If a participant opted to complete the survey online, the survey was sent via email. If the patient chose to complete the survey over the phone, their responses were entered in Opinio by the primary investigator. 

The survey (Appendix A) contained 12 questions. The survey was piloted with kidney transplant patients and hemodialysis patients followed by the Horizon Health Network Nephrology Program. Based on the pilot, the survey was refined prior to submission to the Research Ethics Board. Consent was obtained by completing the survey electronically or verbally when completing the survey over the phone. 

### 2.5. Data Analysis

#### 2.5.1. Quantitative Data

Continuous variables were described using median and interquartile range. Categorical variables were described using frequencies and percentages. The number of drug therapy problems (DTPs) identified by pharmacists per patient, the number of recommendations made by pharmacists per visit, the number of each DTP per patient (unnecessary drug therapy, needs additional drug therapy, ineffective drug therapy, dosage too high, dosage too low, adverse drug reaction, non-adherence) and the number of pharmacist interventions accepted, accepted with change, or not accepted by a Nephrologist per patient following in-person medication reviews and phone medication reviews were compared using a Wilcoxon Signed Rank test (based upon Shapiro–Wilk test for normality; *p* < 0.05 for all variables of interest). An alpha of 0.05 was used for all analyses. All analysis were performed in IBM SPSS Statistics (Version 27, Armonk, NY, USA).

##### Sample Size Estimation

An a priori power analysis using GPower (Version 3.1.9.7) indicated that the above analyses required a minimum sample size of 34 participants (powered for originally intended paired *t*-tests; medium effect size (Cohen’s d): 0.5, alpha: 0.05, power: 0.8, two-tailed test). Medium effect size selected based on the work of Cohen [14].

#### 2.5.2. Qualitative Data

Content analysis was performed to describe the qualitative data from the open-ended questions in the survey. Codes were created inductively as the data was analyzed. Coding was performed by the primary investigator (KA) and reviewed by two co-investigators (LM, HN). 

## 3. Results

### 3.1. Demographics

The goal of the study was to compare the quality of patient care provided by pharmacist telephone medication reviews compared to in-person medication reviews and to measure patient satisfaction with telephone medication reviews in the home dialysis population. Retrospective data collection was conducted on 46 patients that met inclusion criteria, with 84.8% of patients undergoing peritoneal dialysis. Of those 46 patients, 25 (54.3%) completed the patient satisfaction survey. Table 1 provides a summary of participants’ baseline demographics, including relevant comorbidities. The mean participant age was 59.3 years (SD: 12.5; range: 27–81), with a mean time on dialysis of 3.6 years (SD: 2.4; range: 1.4–13.6) (Table 1). Patients were prescribed an average of 11.3 (SD: 3.7; range: 4–20) medications and had an average of 5.9 (SD: 2.3; range 2–12) comorbidities (Table 1). 

### 3.2. Pharmacist-Led Medication Review Outcomes

No difference in the number of recommendations made per visit or the number of total drug therapy problems (DTPs) reported per patient was identified between in-person and telephone visits (*p* > 0.05, Table 2; 102 and 77 total pharmacist recommendations for in-person and telephone visits, respectively). Physicians accepted more pharmacist recommendations per patient from in-person visits compared to telephone visits (*p*
**=** 0.009), though no difference in the number of interventions accepted with change or not accepted was observed (*p* > 0.05, Table 2). Overall, a total of 84 (82.4%) and 45 (58.4%) recommendations were accepted or accepted with change for in-person and telephone visits, respectively.

The types of drug therapy problems (DTPs) recorded for patients during in-person and telephone appointments were identified and were categorized as unnecessary therapy, needed additional drug therapy, ineffective drug therapy, dosage too high, dosage to low, adverse drug reaction and non-adherence (Figure 1, Table 2). No differences were seen in the number of individual types of DTPs identified per patient by pharmacists between in-person and telephone visits (*p* > 0.05 all analyses; Table 2). When seen as proportions within each respective visit type, the category “Patients needing additional therapy” was identified as the most common DTP for both in-person and telephone appointments (31.3% and 24.2%, respectively; Figure 1A,B). Ineffective drug therapy was identified as the least common DTP for in-person and telephone appointments (0.0% and 3.2%, respectively; Figure 1A,B). 

### 3.3. Patient Satisfaction Survey Outcomes

#### 3.3.1. Quantitative Data

Due to the retrospective nature of this study, some of the patients included in the chart review were no longer part of the home dialysis program at the time of the patient satisfaction survey (e.g., they had received a kidney transplant, switched dialysis modality, or were deceased). Twenty-seven of 46 patients were therefore eligible to participate in the satisfaction survey, of which 25 patients consented to participate. Of the 25 participants that completed the patient satisfaction survey, 95.8% agreed or strongly agreed that telephone medication reviews were just as good as in-person medication reviews. All participants agreed or strongly agreed that their privacy and confidentiality were protected and respected and that they were able to communicate effectively with the pharmacist during the telephone medications reviews. One hundred percent of participants agreed or strongly agreed that they were satisfied with the care they received from the dialysis pharmacist on the telephone. When asked how they would prefer medication reviews in the future, 16.7% of participants said they would prefer in-person medication reviews, 20.8% said they would prefer telephone medication reviews, 29.2% said they would prefer an in-person medication review once a year and telephone the rest of the time, while the remaining participants (33.3%) had no preference. 

#### 3.3.2. Qualitative Data

Participants were asked open-ended questions to gather further opinions on telephone medication reviews. The first question asked participants what they would change about telephone medication reviews. Most respondents said that they would change nothing (Table 3). Respondents also indicated that they would like the pharmacist to speak louder, a scheduled appointment time and that they would prefer in-person medication reviews (“I don’t like virtual care”; Table 3). 

The second question asked participants what they liked best about telephone medication reviews. Participants expressed that they liked the convenience and lack of travel with the telephone reviews (Table 3). Respondents also reported that telephone medication reviews had similar quality of care to in-person reviews and that the pharmacists provided efficient and patient centered care (Table 3). One participant said, “There is no difference between in person and telephone visits”, while another participant said, “The pharmacists are open and personable, effective and efficient during every visit.”

When asked to provide additional comments, participants indicated that they were satisfied with the care they received from the pharmacists (Table 3). It was also found that participants perceived better care with in-person visits (Table 3). One participant said, “I feel a disconnect with COVID and telephone visits.” 

## 4. Discussion

Our study provides evidence suggesting that telephone medication reviews by a pharmacist provide similar quality of care to in-person pharmacist medication reviews in home dialysis patients. Within our study cohort, we found no difference in the number of DTPs identified per patient between in-person and telephone medication reviews. Although the number of pharmacist recommendations did not differ between modalities, physician acceptance of pharmacist recommendations was higher with in-person visits. Patients reported a similar quality of care with telephone medication reviews and found them to be convenient and effective. This study supports existing literature that pharmacists have a positive impact on clinical outcomes and maintain clinical services with telephone-based care, with evidence from individual studies and systematic reviews indicating this positive impact existing across clinical care disciplines and in both outpatient and inpatient settings [6,7,8,9,10,15,16].

To our knowledge, this is the first study to compare in-person versus telephone medication reviews by a pharmacist in a dialysis population. Pharmacists identified an average of 2.5 DTPs during in-person medication reviews compared to an average of 2.1 DTPs during telephone reviews (*p* = 0.431). The types of DTPs identified were similar between groups, with the top three categories being “Needed additional therapy” (24.2–31.3%), “Dosage too low” (16.5–18.9%) and “Adverse drug reaction” (13.0–16.8%). 

It is important to highlight that the format and depth of the pharmacist telephone medication reviews were similar to in-person reviews. Implementation of telephone medication reviews led to more robust pharmacist documentation as the interdisciplinary team transitioned from paper documentation to electronic documentation of clinic visits. This was necessary to facilitate communication between team members working remotely, including between pharmacists and nephrologists. Pharmacists communicated their recommendations to the nephrologist through documentation in the electronic chart. The nephrologist would read assessments and recommendations from the pharmacist and other care team members (nursing, dietitian) prior to completing a telephone visit with the patient. Although a standard telephone clinic process was developed over the beginning of the pandemic, physician acceptance of pharmacist recommendations was lower with telephone visits versus in person. This was an unexpected finding. Before the pandemic, in-person visits allowed pharmacists more opportunity to discuss recommendations with nephrologists face-to-face and in real time. We hypothesize that the loss of opportunity for in-person discussions following adoption of a virtual clinic model led to a lower physician acceptance of pharmacist recommendations. This is supported by the findings of a retrospective study of acceptance factors for hospital pharmacist interventions [17]. Pharmacist recommendations were accepted more often when communicated verbally to physicians (either in-person or by phone) rather than by text through a hospital software system (+27.7%, 95% CI: +23.2 to +32.1%) [17]. Alternatively, in a study by Bruns et al. investigating the control of blood pressure, all recommendations made, regardless of appointment type, were accepted by the care team [8]. This difference in findings between studies indicates further work should be performed to identify if acceptance of pharmacist recommendations is based on the type of review and recommendation format, the team dynamic, the clinical care area, or the type of pharmacist intervention/recommendation being made. As a telephone model of care becomes standard practice and pandemic operations normalize, future work should also reinvestigate this finding to identify if physician acceptance rates reach parity between telephone visits and in-person. 

Interestingly, our DTP results differ from prior studies in the general population. A recent retrospective observational cohort study by McNamara et al. of primary care patients found that significantly more DTPs were identified during pharmacist in-person reviews vs. telephone reviews, whereas our study found no difference [18]. Most patient encounters were unique between treatment groups in the McNamara et al. study [18]. Only 26 study patients were reviewed by a pharmacist during both an in-person and telephone visit [18]. In contrast, participants in our study were required to have two in-person medication reviews and two telephone medication reviews to be included. By comparing the same patients between medication review methods, we have mitigated the confounding that may have been introduced by participants themselves. By having some participants present for both visit types and others for only one of the two modalities, confounding may be introduced in studies like McNamara et al. where independence between comparator groups cannot be ensured. This may account for observed differences in DTP findings between our study and theirs. Another possible factor for observed differences may be that McNamara et al. did not follow a standardized process when switching to telehealth given the urgency for change at the beginning of the pandemic [18]. Practices varied from provider to provider and department to department. In contrast, our clinic was able to adopt a standardized telephone clinic process early on due to our smaller program size. It is possible that COVID-19 may have impacted DTPs in our study as patients were less likely to see other health care professionals in-person during the telephone medication review data collection period, however, future work should ultimately look at the long-term drug therapy patient outcomes following the pandemic as this was not a variable of interest for our study. 

Most patients surveyed in our study were satisfied with pharmacist telephone medication reviews. Patients reported a similar quality of care with telephone medication reviews compared to in-person reviews and found them to be convenient and effective. These findings are supported by a survey of 235 non-dialysis CKD patients from Ontario, Canada, who were converted from in-person to telephone visits with their nephrologist in response to the COVID-19 pandemic [19]. In our study, patients were very comfortable with telephone consultation and felt their concerns and preferences were addressed equally well compared with in-person visits. Most patients preferred telephone consultation as it reduced waiting periods, travel time and travel costs. Patients who preferred in-person visits felt that telephone consultation limited interpersonal relationship development with their nephrologist. 

This study has several limitations. First, chart review and classification of DTPs was completed by one reviewer due to limited resources. Having multiple independent reviewers of DTPs would have provided a more robust dataset. Second, the patient satisfaction survey was not validated as there was no existing survey that adequately addressed the goals of this study. The survey was piloted by a small group of transplant and hemodialysis patients for readability, however there is still a risk of response bias with a non-validated survey. It is also recognized that single-centre, retrospective studies on small regional samples inherently limit our ability to generalize our results broadly to health centres in other jurisdictions within or outside Canada. We therefore recommend future studies assess the use of in-person, telephone, or other virtual modalities for pharmacist medication reviews in CKD patients through multi-centre or randomized controlled trial designs to provide strong evidence towards patient care management decisions. Lastly, the survey was offered over the telephone or online, though most respondents chose to complete the survey over the telephone with the primary investigator reading the questions and completing the survey online for the participants. This increases the risk of observation or acquiescence bias as respondents may have been more likely to agree to statements read by an investigator. 

Future work should reassess patient satisfaction to ensure continued acceptance of a telephone-based model of care, especially now that many COVID pandemic restrictions that were in place within healthcare systems during our study have been lifted. Because our study was focused on a pharmacist-led medication review for CKD patients, our main outcomes related to the process measures and the patient satisfaction of such a service. We therefore recommend future work to investigate CKD patient outcomes following in-person or telephone pharmacist-led medication reviews, to provide vital context towards differences in their impact on mortality or morbidity patient outcomes. Patients reported satisfaction with telephone pharmacist care, although not all felt that the telephone could replace in-person visits completely. Future research should focus on quality of pharmacist telephone care and patient outcomes 1–2 years post-pandemic onset to assess whether there are differences in DTPs after care teams are more established in their telephone/virtual care processes. 

## 5. Conclusions

Pharmacists provided similar quality of care to home dialysis patients with telephone medication reviews compared to in-person medication reviews. There was greater physician acceptance of pharmacist recommendations with in-person visits. Overall, patients were satisfied with telephone medication reviews, and most would like to continue to have telephone medication reviews as an option in the future. 

## Figures and Tables

**Figure 1 pharmacy-11-00001-f001:**
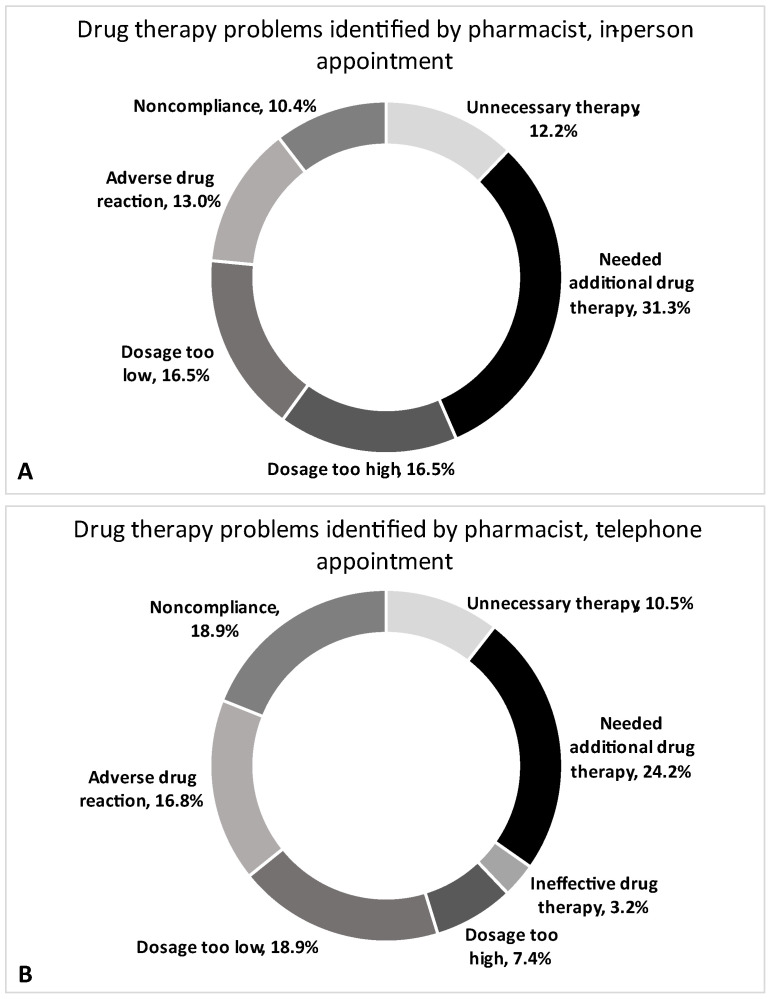
Drug therapy problems (DTPs) identified by pharmacists, during in-person ((**A**), *n* = 115 DTPs) and telephone ((**B**); *n* = 95 DTPs) appointments presented as a proportion of the total DTPs identified.

**Table 1 pharmacy-11-00001-t001:** Baseline characteristics (*n* = 46).

Characteristic	Outcome
**Demographics**	
Male, N (%)	20 (43.5)
Female, N (%)	26 (56.5)
Age, years, mean (SD)	59.3 (12.5)
Time on dialysis, years, mean (SD)	3.6 (2.4)
Number of medications, mean (SD)	11.3 (3.7)
Number of comorbidities, mean (SD)	5.9 (2.3)
**Comorbidities, N (%)**	
Hypertension	39 (84.8)
Diabetes	25 (54.3)
Dyslipidemia	16 (34.8)
Osteoarthritis	8 (17.4)
Gout	6 (13.0)
Atrial Fibrillation	5 (10.9)
Polycystic Kidney Disease	5 (10.9)
GERD	4 (8.7)
COPD	4 (8.7)
Peripheral Vascular Disease	4 (8.7)

**Table 2 pharmacy-11-00001-t002:** Comparison of pharmacist interventions between in-person and telephone modalities.

Measure	Median (IQR)	*p*-Value	ES
Number of recommendations, per visit		0.200	−0.13
In-person	1.0 (0.5–1.5)		
Telephone	1.0 (0.38–1.5)		
**Acceptance of pharmacist intervention**		**0.009**	**−0.27**
**In-person**	**1.0 (1.0–2.0)**		
**Telephone**	**0.0 (0.0–2.0)**		
No acceptance of pharmacist intervention		0.057	0.20
In-person	0.0 (0.0–1.0)		
Telephone	0.0 (0.0–1.0)		
Acceptance of pharmacist intervention with change		0.564	−0.06
In-person	0.0 (0.0–0.0)		
Telephone	0.0 (0.0–0.0)		
Total drug therapy problems (DTPs)		0.431	−0.08
In-person	2.0 (1.0–4.0)		
Telephone	2.0 (1.0–3.0)		
Unnecessary drug therapy		0.415	−0.09
In-person	0.0 (0.0–0.25)		
Telephone	0.0 (0.0–0.0)		
Needed additional drug therapy		0.094	−0.17
In-person	1.0 (0.0–1.0)		
Telephone	0.0 (0.0–1.0)		
Ineffective drug therapy		0.083	0.18
In-person	0.0 (0.0–0.0)		
Telephone	0.0 (0.0–0.0)		
Dosage too high		0.089	−0.18
In-person	0.0 (0.0–1.0)		
Telephone	0.0 (0.0–0.0)		
Dosage too low		0.968	−0.004
In-person	0.0 (0.0–1.0)		
Telephone	0.0 (0.0–1.0)		
Adverse drug reaction		0.909	0.01
In-person	0.0 (0.0–0.25)		
Telephone	0.0 (0.0–1.0)		
Non-adherence		0.186	0.14
In-person	0.0 (0.0–0.0)		
Telephone	0.0 (0.0–1.0)		

*n* = 46; Wilcoxon signed rank test; IQR: Interquartile range represented as 25th–75th percentile; ES: Effect size, represented in the form of r. Bold denotes significant difference between groups. Values are presented per patient, unless otherwise stated.

**Table 3 pharmacy-11-00001-t003:** Content analysis of open-ended questions in patient satisfaction survey.

What Would You Change about the Telephone Medication Reviews with the Pharmacist?		
Theme	# of Statements	Quotes
In-person visits only	1	“I don’t like virtual care”
Schedule appointment times in advance	1	“The one thing I would prefer would be a designated time and date, as opposed to getting an unexpected call. I like to have my med list in front of me, and have any questions I may have, prepared.”
Speak Louder	1	“If the pharmacist spoke a little louder.”
Change Nothing	16	“I wouldn’t change anything”
**What do you Like Best about the Telephone Medication Reviews with the Pharmacist?**		
**Theme**	**# of Statements**	**Quotes**
Similar quality of care	3	“There is no difference between in person and telephone visits”
No travel required	7	“I live out of town, saves having to go for in person visit” “I don’t have to go to the hospital”
Efficient and patient-centered care	5	“The pharmacists are open and personable, effective and efficient during every visit.” “The professional yet personal experience”
Convenient for patient	7	“Quick, easy, and it saves time” “More convenient, I don’t have to be there” “It doesn’t take as much time”
Enjoy all of it	1	“All of it”
**Please Provide any Additional Comments**		
**Theme**	**# of Statements**	**Quotes**
Patients are satisfied with the care they receive from pharmacists	5	“I am satisfied with the care I receive from the pharmacists” “I find the pharmacists to be very thorough, professional and informative.”
Better care is perceived to be obtained through in-person visits	3	“I feel if they see me they will do better, but you can’t do that on the phone. If you don’t tell them everything that’s going on they won’t know.” “I feel a disconnect with COVID and telephone visits.”
No issues	2	“No issues with pharmacy”

## Data Availability

Due to privacy and confidentiality policies and regulations, data from this study cannot be shared outside of study investigators.

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
