# Peer review of "Telephone versus In-Person Pharmacist-Led Medication Reviews in Home Dialysis Patients: Evaluating Quality of Care and Patient Satisfaction"

_pharmacy, 2022, doi:10.3390/pharmacy11010001_

Round 1
Reviewer 1 Report
First of all, I would like to congratulate the authors of the article, they wrote an interesting article which compares the quality of care telephone medication review vs in person review.
Minor changes:
I recommend indicating the number of patients in the abstract.
I recommend to indicate if sample size calculation. If it is not calculated, I recommend indicating it in the limitations.
I recommend clarifying the number of people who refused to participate in the study and the reasons, as well as the reasons of 46 satisfaction surveys were not obtained and only 25 were obtained.
Line 219: It cannot be stated in the text that the relationship has been “demonstrated”, since the sample size is very small and it is possible that it does not detect significant differences due to the low power of the study. I suggest changing the word "demonstrated" to "hypothesized". On the other hand, descriptive studies without a comparator group cannot demonstrate causality.
In limitations, I would suggest that future works should be done as clinical trials design with an adequate sample size should to demonstrated obtained results.
Author Response
Dear editor,
We thank the reviewers for their thoughtful comments on our manuscript “Telephone versus in-person pharmacist-led medication reviews in home dialysis patients: evaluating quality of care and patient satisfaction”. We have addressed the comments to the best of our ability in the time allotted and have provided responses below in addition to our revised manuscript. Below, the reviewer comments are in bold, separated by reviewer, with our response following each comment. For the revised manuscript, we have retained the track changes as requested in your email, but have also set the text to red where changes have been made should you or the reviewers prefer to read the document with no mark-up but still wish to see where the revisions have been made.
Thank you.
Reviewer 1
Minor changes:
I recommend indicating the number of patients in the abstract.
We have added the number of patients to the abstract.
I recommend to indicate if sample size calculation. If it is not calculated, I recommend indicating it in the limitations.
The sample size calculation was included in the quantitative data analysis plan. We have created its own section header to indicate its location and added further details to explain the calculation more clearly. No separate sample size calculation for the survey was performed as the participation only extended to those currently in dialysis care at our regional centre.
I recommend clarifying the number of people who refused to participate in the study and the reasons, as well as the reasons of 46 satisfaction surveys were not obtained and only 25 were obtained.
Due to the retrospective nature of this study, some of the patients included in the chart review were no longer part of the home dialysis program at the time of the patient satisfaction survey (e.g., they had received a kidney transplant, switched dialysis modality, or were deceased). They were therefore not eligible to participate in the survey. This has been clarified in the results.
Line 219: It cannot be stated in the text that the relationship has been “demonstrated”, since the sample size is very small and it is possible that it does not detect significant differences due to the low power of the study. I suggest changing the word "demonstrated" to "hypothesized". On the other hand, descriptive studies without a comparator group cannot demonstrate causality.
“Demonstrated” has been changed to “provides evidence suggesting”. As described in the manuscript methodology, our power analysis using GPower (version 3.1.9.7) indicated that this study required a minimum sample size of 34 participants (powered for paired t-tests; medium effect size (Cohen’s d): 0.5, alpha: 0.05, power: 0.8, two-tailed test). We met this sample size with n=46 patients.
In limitations, I would suggest that future works should be done as clinical trials design with an adequate sample size should to demonstrated obtained results.
The following sentence has been added to the limitations paragraph to address this comment: “It is also recognized that single-centre, retrospective studies on small regional samples inherently limit our ability to generalize our results broadly to health centres in other jurisdictions within or outside Canada. We therefore recommend future studies assess the use of in-person, telephone, or other virtual modalities for pharmacist medication reviews in CKD patients through multi-centre or randomized controlled trial designs to provide strong evidence towards patient care management decisions”.
Reviewer 2 Report
This is a nice self-contained service evaluation with one component changed - mode of delivery, providing an interesting research question
Within the introduction it would be helpful to cite general literature regarding telephone versus face to face service delivery - there is an expanding literature regarding this which is not captured here. It would be good to know that this was used to inform the creation of the survey.
It is usual for training to be provided to prepare healthcare professionals for the delivery of services via telephone rather than face to face. Was any training undertaken here? This is helpful to understand patient satisfaction and what could have been done to enhance patient experience.
Is the clinical dialysis pharmacist before and after the same - this needs stating as this is a potentially confounding variable
Is the clinical dialysis pharmacist the researcher who performed the analysis because again this could affect objectivity within analysis
Could the potential for DTPs be affected by COVID i.e. if patients were less likely to see other healthcare professionals face to face could this increase the likelihood of DTPs, thereby increasing the opportunity for DTPs? This would affect comparability of the two periods.
Analysis was based on patients being their own controls i.e. - were they the same patients in both arms? If this was the case then surely the DTPs identified prior to the service being via telephone would reduce the DTPs after? Would this also partially explain why there were less recommendations after the switch?
Acceptance should be calculated as a proportion of recommendations not number accepted and then a chi-squared analysis used i.e. McNemar's test
The authors have performed Wilcoxon rank test assuming the 'differences' before and after were not normally distributed - is there any evidence to support this?
If a non-parametric test was used (Wilcoxon), why was the sample size calculation based on a paired t-test?
We only have the power to detect a medium effect size - what sample size would be required for a small effect size? I fear that there is an effect but there is type 2 error throughout.
Table 1 would be better if it compared questionnaire respondents with non-respondents within it.
Table 2 requires review - either use Median or Mean - not both- The data do look to be normally distributed for number of recommendations
Other comparisons in table 2 should be proportions not numbers
Table 2 (second one -error) again mean or median and no differences seen
Author switches between adherence and compliance - can we please use the former
Rather than table 2 (the second table 2- error) and the myriad of pie charts - which repeat data in table 3 - this could all be one bar chart
Minor point:
Patients are not 'on' medicines, they are 'prescribed' them - 'on' suggests prescribed and taking which we dont know
There is usually only one primary outcome - section 3.2 is Process Outcomes
We do not need figures A to E as each essentially provides one or two pieces of data. Again perhaps better as a figure
Interesting presentation of qualitative data with counting of frequency of codes - would be better to be presented as thematic analysis rather than counting how often something is said.
I think the discussion needs rewriting as it suggest no differences when there was insufficient power to state that small differences may have occurred. Stats tests and data presentation is not ideal and once this is corrected the discussion requires review, with a significant section acknowledging limitations.
The focus on DTPs is not ideal as this is a process measure generated by the HCP providing the service - we should be more focused on patient outcomes.
The literature does not state that pharmacists have a positive impact - it states that the evidence is poor quality and better quality research is required. i.e., RCTs not service evaluations.
Author Response
Dear editor,
We thank the reviewers for their thoughtful comments on our manuscript “Telephone versus in-person pharmacist-led medication reviews in home dialysis patients: evaluating quality of care and patient satisfaction”. We have addressed the comments to the best of our ability in the time allotted and have provided responses below in addition to our revised manuscript. Below, the reviewer comments are in bold, separated by reviewer, with our response following each comment. For the revised manuscript, we have retained the track changes as requested in your email, but have also set the text to red where changes have been made should you or the reviewers prefer to read the document with no mark-up but still wish to see where the revisions have been made.
Thank you.
Reviewer 2
This is a nice self-contained service evaluation with one component changed - mode of delivery, providing an interesting research question
Within the introduction it would be helpful to cite general literature regarding telephone versus face to face service delivery - there is an expanding literature regarding this which is not captured here. It would be good to know that this was used to inform the creation of the survey.
A sentence has been added to the introduction to address this comment.
It is usual for training to be provided to prepare healthcare professionals for the delivery of services via telephone rather than face to face. Was any training undertaken here? This is helpful to understand patient satisfaction and what could have been done to enhance patient experience.
We have added additional info in methods section (Pharmacist Medication reviews section) to address this.
Is the clinical dialysis pharmacist before and after the same - this needs stating as this is a potentially confounding variable.
Yes – same group of pharmacists. We have clarified this point in the methods.
Is the clinical dialysis pharmacist the researcher who performed the analysis because again this could affect objectivity within analysis.
No, these researchers did not participate in patient care. Data collection was performed by a pharmacist outside of the circle of care (KA) with quantitative data analysis being led by a non-pharmacist researcher (AJF). We have added information to the methods to clarify this position.
Could the potential for DTPs be affected by COVID i.e. if patients were less likely to see other healthcare professionals face to face could this increase the likelihood of DTPs, thereby increasing the opportunity for DTPs? This would affect comparability of the two periods.
The following line has been added to the discussion: “It is possible that COVID-19 may have impacted DTPs in our study as patients were less likely to see other health care professionals in-person during the telephone medication re-view data collection period, however, future work should ultimately look at the long-term drug therapy patient outcomes following the pandemic as this was not a variable of inter-est for our study.”
Analysis was based on patients being their own controls i.e. - were they the same patients in both arms? If this was the case then surely the DTPs identified prior to the service being via telephone would reduce the DTPs after? Would this also partially explain why there were less recommendations after the switch?
Yes, these were the same patients in both groups. If the in-person med review had been the patient’s very first medication review, we would expect more DTPs to be identified on first review. However, patients were reviewed by a pharmacist 4 times a year prior to switching to telephone medication reviews. I.e. the first in-person medication review used in this study was not the first review by a pharmacist (for most patients), therefore we don’t believe this would have contributed to a difference in DTPs. Patients who had been on home dialysis for >1 year prior to study onset would have had 4+ medication reviews with a pharmacist (at q 3 month intervals) prior to switching to a telephone format. We have added some clarification in the methods regarding the pharmacist medication review format.
Acceptance should be calculated as a proportion of recommendations not number accepted and then a chi-squared analysis used i.e. McNemar's test.
We certainly agree that proportions of outcome would be a valuable mode of data presentation for a descriptive/exploratory study; however, due to the nature of the variables of interest being counts of occurrence per patient or per visit, and the repeated measures design used, we would be unable to analyze our data using proportions without incurring confounding in our results due to a failure to meet the assumption of independence of observations or mutual exclusivity (should either a chi-square test or McNemar test be used). We therefore opted to present the data as rates per patient or visit to best represent on average the experience of the patient and pharmacist interacting through their in-person or telephone meetings.
We have added appropriate proportions where possible for descriptive purposes.
The authors have performed Wilcoxon rank test assuming the 'differences' before and after were not normally distributed - is there any evidence to support this?
A Shapiro-Wilk test was performed prior to analysis of the data. A paired t-test was the intended analysis, with the Wilcoxon test used following the results of the test for normality. Details for this have been added to the data analysis plan.
If a non-parametric test was used (Wilcoxon), why was the sample size calculation based on a paired t-test?
As the power analysis was performed a priori, we had originally intended to perform a paired t-test and subsequently performed our power analysis and sample size estimation with that analysis in mind.
We only have the power to detect a medium effect size - what sample size would be required for a small effect size? I fear that there is an effect but there is type 2 error throughout.
An a priori power analysis was performed for this study using a medium effect size. We certainly recognize that the choice of the effect size in this estimation can greatly impact the sample size indicated for a research study (i.e., smaller effect sizes used in estimation result in larger sample sizes required). Without a clear consensus in an area of research prior to the commencement of a research study as to the effect size and its potential relationship to a clinical effect, there can be difficulty in selecting an appropriate effect size for sample size estimation/power analysis. According to Cohen (1988), a medium effect size (d = 0.5) is “one large enough to be visible to the naked eye.” Powering the study for a smaller effect size, perhaps Cohen’s d = 0.2, may lead to a reduced chance of committing a type 2 statistical error, if one is present, but may also lead to a discussion surrounding small differences that don’t translate to any meaningful clinical impact. We feel that the medium effect size used is a compromise between identifying a clinically relevant effect and a reasonable minimization of any type 2 errors present.
Cohen, J. Statistical power analysis for the behavioral sciences, 2nd ed.; Lawrence Erlbaum Associates, Publishers: 1988.
Table 1 would be better if it compared questionnaire respondents with non-respondents within it.
We did not differentiate demographic data by questionnaire respondent vs. non-respondent during data collection, therefore we are unable to separate the data this way. Also, due to the retrospective nature of the study, not all patients included in the retrospective chart review were eligible/included in the patient questionnaire. (e.g. some were deceased, had received a transplant, or switched dialysis modality).
Table 2 requires review - either use Median or Mean - not both- The data do look to be normally distributed for number of recommendations.
We have adjusted the tables and figures accordingly to represent the data more accurately. We have only included the median and interquartile range in table 2.
Other comparisons in table 2 should be proportions not numbers
We certainly agree that proportions of outcome would be a valuable mode of data presentation for a descriptive/exploratory study; however, due to the nature of the variables of interest being counts of occurrence per patient or per visit, and the repeated measures design used, we would be unable to analyze our data using proportions without incurring confounding in our results due to a failure to meet the assumption of independence of observations or mutual exclusivity (should either a chi-square test or McNemar test be used). We therefore opted to present the data as rates per patient or visit to best represent on average the experience of the patient and pharmacist interacting through their in-person or telephone meetings.
We have added appropriate proportions where possible for descriptive purposes.
Table 2 (second one -error) again mean or median and no differences seen
Table 2 has been updated to only include the median and interquartile range as a reflection of the non-parametric test that was performed.
Author switches between adherence and compliance - can we please use the former
“Compliance” has been changed to “adherence” throughout manuscript.
Rather than table 2 (the second table 2- error) and the myriad of pie charts - which repeat data in table 3 - this could all be one bar chart.
We have removed figure 2 to reduce the number of pie charts in the manuscript and restricted this information to the text of the result section. We have simplified table 2 to only contain the median and interquartile range. To easily present the results, including the results of our Wilcoxon signed rank tests and the effect sizes observed, we have opted to retain table 2 as a series of box plots were not amenable to the presentation of our results.
Minor point:
Patients are not 'on' medicines, they are 'prescribed' them - 'on' suggests prescribed and taking which we dont know.
We have corrected this language from “Patients were on an average of 11.3 (SD: 3.7; range: 4-20) medications” to “Patients were prescribed an average of 11.3 (SD: 3.7; range: 4-20) medications”.
There is usually only one primary outcome - section 3.2 is Process Outcomes.
We have updated the section headings to distinguish the two parts of our study, “pharmacist-led medication review outcomes” & “patient satisfaction survey outcomes”.
We do not need figures A to E as each essentially provides one or two pieces of data. Again perhaps better as a figure
Figure 2 has been removed from the manuscript.
Interesting presentation of qualitative data with counting of frequency of codes - would be better to be presented as thematic analysis rather than counting how often something is said.
Given the design of the study and the format in which the data was collected (i.e., through open-ended survey questions), a thematic analysis would not be possible at this time. We feel that a content analysis is best applied in this context due to the depth of the questions being asked and aims for the patient satisfaction survey.
I think the discussion needs rewriting as it suggest no differences when there was insufficient power to state that small differences may have occurred. Stats tests and data presentation is not ideal and once this is corrected the discussion requires review, with a significant section acknowledging limitations.
The following sentence was added to the discussion: “It is also recognized that single-centre, retrospective studies on small regional samples inherently limit our ability to generalize our results broadly to health centres in other jurisdictions within or outside Canada. We therefore recommend future studies assess the use of in-person, telephone, or other virtual modalities for pharmacist medication reviews in CKD patients through multi-centre or randomized controlled trial designs to provide strong evidence towards patient care management decisions”.
As described above, we performed an a priori power analysis based on a relevant effect size and ultimately exceeded the minimum sample size identified by our power analysis. We do recognize that there still exists a chance for a type 2 error, if one is present, and have therefore altered the language used in our discussion and included additional language in our limitations section of this manuscript.
The focus on DTPs is not ideal as this is a process measure generated by the HCP providing the service - we should be more focused on patient outcomes.
This is a good point. The current study was designed to compare outcomes of a process change. It was powered to detect differences in the number of DTPs and physician acceptance of pharmacist recommendations. We chose these outcomes as they have been used previously in pharmacy research. Assessing patient outcomes is an excellent next step, however, this will require a much larger sample size. Our study was not powered for this. This would be a great direction for future research. A sentence has been added to the discussion stating the same.
The literature does not state that pharmacists have a positive impact - it states that the evidence is poor quality and better quality research is required. i.e., RCTs not service evaluations.
The wrong reference was listed here. We have corrected this to reference the 2018 Niznik/Kane-Gill study which supports that Clinical pharmacy telemedicine interventions in the outpatient or ambulatory setting, primarily via phone, have an overall positive impact on outcomes related to clinical disease management, patient self-management, and adherence in the management of chronic diseases.
Round 2
Reviewer 2 Report
Happy with nearly all changes and responses.
I would however like to see a paragraph of text citing the wider literature regarding the move from face to face to online consultations in the introduction. The authors have not addressed this adequately. In fact I could not see the sentence.
This literature should then be revisited in the discussion.
Author Response
Happy with nearly all changes and responses.
I would however like to see a paragraph of text citing the wider literature regarding the move from face to face to online consultations in the introduction. The authors have not addressed this adequately. In fact I could not see the sentence.
Beginning on line 41 of page 1 (with all markups shown), the following sentences have been added to the second paragraph of the introduction:
The COVID-19 pandemic introduced unprecedented challenges to direct patient care in the ambulatory setting. Telepharmacy gained popularity as a method of supporting pharmacist medication management while maintaining social distancing [5]. A pre-COVID systematic review of 34 studies demonstrated that clinical pharmacy tele-medicine interventions in the outpatient or ambulatory setting, primarily via phone, had an overall positive impact on chronic disease management [6], a finding corroborated by a systematic review of inpatient ICU and non-ICU telepharmacy services published in the same year [7]. The evidence supporting telepharmacy in the ambulatory care setting has continued to grow since the COVID-19 pandemic, with studies showing benefit of tele-pharmacy interventions in chronic diseases such as hypertension [8] and diabetes [9,10] and in the management of anticoagulation medications [11]. Unfortunately, there are no published studies of telepharmacy interventions in chronic kidney disease (CKD) or dialysis patients to date.
This literature should then be revisited in the discussion.
We have re-worked the first paragraph of the discussion to tie in our overall findings with those found in the literature by moving a sentence from the final paragraph ahead (beginning line 280, page 10 – with all markups shown). As such, we have reworked the end of the discussion to allow for this change (beginning line 372, page 11). The following references have been added to the discussion (line 284):
- Cao DX, Tran RJC, Yamzon J, Stewart TL, Hernandez EA. Effectiveness of telepharmacy diabetes services: A systematic review and meta-analysis. Am J Health Syst Pharm. 2022 May 24;79(11):860-872.
- Bruns BE, Lorenzo-Castro SA, Hale GM. Controlling Blood Pressure During a Pandemic: The Impact of Telepharmacy for Primary Care Patients. J Pharm Pract. 2022 Oct 27:8971900221136629. doi: 10.1177/08971900221136629. Epub ahead of print.
- Strnad, K.; Shoulders, B.R.; Smithburger, P.L.; Kane-Gill, S.L. A systematic review of ICU and non-ICU clinical pharmacy services using telepharmacy. Ann. Pharmacother. 2018, 52, 1250-1258, doi:10.1177/1060028018787213.
The post-COVID McNamara paper remains the key study in our discussion as their methodology was similar to our study. It remains the best fit for contextualizing results. Many of the newer post-COVID studies used different methodology and outcome measures, therefore making direct comparison difficult. We have therefore left the core of our discussion unchanged while acknowledging the overall positive impact of telepharmacy at the beginning of our discussion.